# Relative COVID-19 Viral Persistence and Antibody Kinetics

**DOI:** 10.3390/pathogens10060752

**Published:** 2021-06-13

**Authors:** Chung-Guei Huang, Avijit Dutta, Ching-Tai Huang, Pi-Yueh Chang, Mei-Jen Hsiao, Yu-Chia Hsieh, Shu-Min Lin, Shin-Ru Shih, Kuo-Chien Tsao, Cheng-Ta Yang

**Affiliations:** 1Department of Laboratory Medicine, Chang Gung Memorial Hospital, Taoyuan 33333, Taiwan; joyce@cgmh.org.tw (C.-G.H.); changpy@cgmh.org.tw (P.-Y.C.); m9205025@stmail.cgu.edu.tw (M.-J.H.); srshih@mail.cgu.edu.tw (S.-R.S.); kctsao@adm.cgmh.org.tw (K.-C.T.); 2Department of Medical Biotechnology and Laboratory Science, College of Medicine, Chang Gung University, Taoyuan 33302, Taiwan; 3Research Center for Emerging Viral Infections, College of Medicine, Chang Gung University, Taoyuan 33302, Taiwan; duttijiva@gmail.com; 4Division of Infectious Diseases, Department of Medicine, Chang Gung Memorial Hospital, Taoyuan 33333, Taiwan; chingtaihuang@gmail.com; 5Division of Infectious Diseases, Department of Medicine, College of Medicine, Chang Gung University, Taoyuan 33302, Taiwan; 6Division of Infectious Diseases, Department of Pediatrics, Chang Gung Memorial Hospital, Taoyuan 33333, Taiwan; yuchiahsieh@gmail.com; 7Division of Infectious Diseases, Department of Pediatrics, College of Medicine, Chang Gung University, Taoyuan 33302, Taiwan; 8Department of Thoracic Medicine, Chang Gung Memorial Hospital, Taoyuan 33333, Taiwan; smlin100@gmail.com; 9Department of Respiratory Therapy, College of Medicine, Chang Gung University, Taoyuan 33302, Taiwan

**Keywords:** COVID-19, viral persistence, serum antibody, neutralization efficacy, cytokine profile

## Abstract

A total of 15 RT-PCR confirmed COVID-19 patients were admitted to our hospital during the in-itial outbreak in Taiwan. The average time of virus clearance was delayed in seven patients, 24.14 ± 4.33 days compared to 10.25 ± 0.56 days post-symptom onset (PSO) in the other eight pa-tients. There was strong antibody response in patients with viral persistence at the pharynx, with peak values of serum antibody 677.2 ± 217.8 vs. 76.70 ± 32.11 in patients with delayed versus rapid virus clearance. The patients with delayed viral clearance had excessive antibodies of compromised quality in an early stage with the delay in peak virus neutralization efficacy, 34.14 ± 7.15 versus 12.50 ± 2.35 days PSO in patients with rapid virus clearance. Weak antibody re-sponse of patients with rapid viral clearance was also effective, with substantial and comparable neutralization efficacy, 35.70 ± 8.78 versus 41.37 ± 11.49 of patients with delayed virus clearance. Human Cytokine 48-Plex Screening of the serial sera samples revealed elevated concentrations of proinflammatory cytokines and chemokines in a deceased patient with delayed virus clear-ance and severe disease. The levels were comparatively less in the other two patients who suf-fered from severe disease but eventually survived.

## 1. Introduction

In late January 2020, we started to treat real-time polymerase chain reaction (RT-PCR)-confirmed COVID-19 patients. We are a medical center that typically cares for patients with moderate to severe diseases. Because of the low prevalence of COVID-19 in Taiwan and local government policy, however, we also admitted COVID-19 patients with mild disease or even those without symptoms for inpatient care. Serial RT-PCR tracking of pharyngeal samples was performed throughout each patient’s hospital course. With informed consent from patients or their families, our research was conducted using serum samples remaining after routine medical tests. We used two ELISA-based kits [1,2,3] to detect anti-spike protein IgG antibodies, and the results of the two were concordant. The tests were semi-quantitative and measured antibody concentrations relative to a cut-off point value in serial dilutions of serum samples. We cultured virus strains from our patient samples and used one of the strains to quantify the neutralization valence of serum samples in our Biosafety Level-3 laboratory. We also measured cytokines and chemokines in the serial sera samples of four patients using the Bio-Plex Pro Human Cytokine 48-Plex Screening kit from Bio-Rad. By mid-March 2020, we had collected serum samples from 15 consecutive patients. As this was a single-center study, we also collected a complete medical record including detailed travel, occupation, contacts, and cluster history.

## 2. Results

### 2.1. Relative Viral Persistence

A total of 15 RT-PCR confirmed COVID-19 patients were admitted to the Linkou campus of Chang Gung Memorial Hospital during the study period as a result of the initial outbreak in Taiwan. All 15 COVID-19 patients were included in this study (Table 1). They suffered from a range of asymptomatic to severe diseases, and virus clearance varied from day 7 to day 49 in these patients (Figure 1). We divided our 15 patients into 2 groups according to whether they may clear the virus within 2 weeks post-symptom onset (PSO). Of the participants involved, 7 patients cleared the virus after two weeks, and 8 patients eradicated the virus within two weeks. For the 7 patients with delayed clearance, the average time of virus clearance was 24.14 ± 4.33 days PSO, and for the 8 patients with rapid clearance, the average time to clear the virus was 10.25 ± 0.56 days PSO (*p* = 0.0046; Figure 2A). There was a significant difference in the age of the two groups of patients. Older patients could not eradicate the virus in time (60.14 ± 3.58 vs. 38.25 ± 5.28 years, *p* = 0.0054). The slope of the virus decline was flat in patients with delayed clearance, in contrast to a sharp slope of virus decline in patients with rapid clearance. We then used the area under curve (AUC) analysis to compare the effective existence of the virus. There was 52% more area under the curve for patients with delayed clearance compared to patients with rapid clearance. The Ct values of E gene at disease presentation were comparable between both the groups of patients (26.02 ± 3.15, *n* = 7 vs. 25.17 ± 4.57, *n* = 8, *p* = 0.693; Figure 2A). All these differences between the two groups were not associated with the initial virus burden at disease presentation.

### 2.2. Antibody Kinetics and Relative Viral Persistence

The serum antibody levels were much higher in patients with delayed virus clearance. The peak values of serum antibodies were 677.2 ± 217.8 vs. 76.70 ± 32.11 in patients with delayed versus rapid virus clearance (*p* < 0.0001; Figure 2B). With comparable levels of initial antibody response, the peak antibody response also emerged later in the patients with delayed virus clearance. The time from symptom onset to the time of peak serum antibody levels was 17.43 ± 2.61 days PSO and 11.13 ± 2.48 days PSO in delayed and rapid clearance groups, respectively (*p* = 0.0004; Figure 2B). 

We then used a neutralization test to evaluate the quality of antibodies with two assessments, neutralizing capacity per unit serum and neutralization efficacy per unit antibody. The capacity was higher in the delayed clearance group (168.00 ± 63.42 vs. 29.68 ± 14.82, *p* = 0.0413; Figure 2C). However, the neutralization efficacy per unit antibody was comparable between the delayed and the rapid clearance groups (41.37 ± 11.49 and 35.70 ± 8.78; *p* = 0.6975; Figure 2D). It is also interesting that the time for peak neutralization efficacy was significantly longer in the delayed clearance group (34.14 ± 7.15 vs. 12.50 ± 2.35 days PSO, *p* = 0.0094; Figure 2E). As the patients in the delayed clearance group had a huge antibody response at first, there were many more antibodies of compromised efficacy before the time of peak efficacy, shown as the shaded area, compared to the patients in the rapid clearance group (Figure 2F). 

Of the participants involved, 2 of the 15 patients died. The deceased had a higher viral load at presentation and a larger amount of poor quality antibodies. Strong but poor-quality antibody response, probably associated with delayed clearance of the virus, was a factor for the less favorable clinical outcome of the disease (Figure 2G).

### 2.3. Inflammation and Relative Viral Persistence

Only four patients in our cohort required ICU assistance. One of them belongs to the group of rapid virus clearance. The patient had other comorbidity and succumbed suddenly after a brief hospital stay. We thus do not have serial serum samples of this patient. We also do not have serial serum samples of the patients with rapid virus clearance and mild disease symptoms, owing to their brief hospital stay and lower number of follow-up tests. Only one such patient donated blood over a follow-up period of two months. We found that the levels of inflammatory cytokines and chemokines were higher in the three patients with delayed virus clearance and severe disease than in the asymptomatic patient with rapid virus clearance. Among patients with relative virus persistence and severe disease, the deceased had high levels of inflammatory cytokines including IFN-γ, IL-17A, IL-6, LIF (leukemia inhibitory factor of IL-6 family), IL-2, IL-3, IL-16, IL-18, and M-CSF (closed squares, Figure 3A), as well as inflammatory chemokines CXCL-9, CXCL-10 (IP-10), CCL-2, and CCL-7 (closed squares, Figure 3B). The stem cell factor (SCF) has previously been linked with airway inflammation [4,5], and the level of SCF was high in serial sera samples of the deceased patient (closed squares, Figure 3C). The deceased had a high level of immune activation-associated molecule IL-2RA (CD25) as well (closed squares, Figure 3D). For the other two patients with severe disease who survived, high levels of IL-12p70, IL-13, CXCL-9, and IL-12RA were present in one of the two (closed circles, Figure 3). There were minimal levels of proinflammatory cytokines and chemokines in the other patient with a disease of less severity (closed triangles, Figure 3). The asymptomatic patient with rapid virus clearance had detectable levels of IFN-γ, IL-6, LIF, IL-16, IL-18, M-CSF, CXCL-9, SCF, and IL-12RA only on day 7 PSO during two-month follow-up (open circles, Figure 3). Interestingly, IL-10 was always higher in this patient, compared to the patients with severe disease (open circles, Figure 3E). 

## 3. Discussion

Our results show strong antibody response in patients with relative viral persistence at the pharynx. They had excessive antibodies of compromised quality in an early stage with the delay in peak of virus neutralization efficacy per unit of antibody. Weak antibody response of patients with rapid viral clearance was also effective, with substantial and comparable neutralization efficacy. Viral persistence boosted inflammatory immune activation. Among patients with delayed virus clearance and ICU assistance, concentrations of proinflammatory cytokines and chemokines were higher in the deceased patient than that in the patients who suffered from severe disease but eventually survived.

Strong antibody response, in terms of high antibody level and proportionally high neutralization titer in the sera, with slower clearance of the SARS-CoV-2 virus has been reported in the literature [6,7,8,9]. The antibody levels began to decline two weeks PSO [10,11]. We also observed strong antibody responses in our patients with relative virus persistence, and the antibody levels started to decline two weeks PSO. Despite this decline in antibody level, we found that neutralization efficacy per unit of antibody remained the same or continued to increase in these patients. This indicates that the proportion of antibodies with lower neutralization efficacy gradually decreases, while the proportion of higher efficacy gradually increases with time. The phenomenon of neutralization efficacy increasing over time is in line with the known maturation process of the antibody response. Created through random VDJ recombination, the B cell receptor (BCR) repertoire is highly heterogeneous. Clonal selection is achieved through stimulation and response where B cells with BCR and antibodies of effective neutralization ability gradually expand and become the major B cell pool responding to the virus. Neutralization represents the antibody’s ability to protect against specific pathogens. It deserves special attention because there is a population of antibodies with poor neutralization capacity in the early stages of the antibody reaction. One of the most concerning risks of convalescent plasma therapy for COVID-19 is that some plasma antibodies may in fact not be protective [12]. They could even be harmful due to mechanisms such as antibody-dependent enhancement (ADE) [13]. Therefore, we must be cautious about the timing of plasma procurement from patients who have recovered from the illness.

People tend to try to link the association between the amounts of virus in respiratory samples and the severity of illness [14,15]. However, the persistent presence of the virus rather than the absolute amount of virus at the throat was responsible for a strong and early antibody response in our cohort of COVID-19 patients. A strong and early antibody response likely predominantly comprises less protective and potentially even deleterious antibodies. In patients with SARS, it was reported that poor clinical outcomes were associated with the early appearance of antibodies [16]. Patients with difficulty eradicating the virus suffer from the damage caused by both the virus and the ineffective potentially deleterious antibodies. In our study, patients with viral persistence and an earlier and stronger antibody response tended to be older. This may explain the vulnerability to COVID-19 in the elderly.

In our observations, low or even no detectable antibodies did not necessarily represent an absence of immunity. Although the absolute antibody quantity in these patients is low, the neutralization efficacy per unit of antibody is equivalent to that of the group with higher antibody levels, indicating that patients with low antibody quantities also have a considerable number of mature B cells secreting effective antibodies. Upon subsequent encounters with the virus, these B cells will likely expand with a memory response and may produce effective antibodies in quantities sufficient to protect the host.

More and more evidence indicates a hyperinflammatory response to SARS-CoV-2 contributes to the development of ARDS, disease severity, and death in COVID-19 [17,18,19,20,21,22]. We also detected high levels of proinflammatory cytokines and chemokines in patients with viral persistence and severe disease requiring intensive care. This trend was much more exaggerated in the deceased. The deceased patient had elevated IL-6 with an increase in other cytokine and chemokine levels in the serum, also similar to the reported literature [19,23,24,25]. There was persistence of elevated levels of IFN-γ IL-17A, IL-6 family member LIF, and many other pro-inflammatory cytokines and chemokines in this deceased patient, compared to those in patients who had delayed virus clearance and suffered from severe disease but eventually survived. Interestingly, there was a high IL-10 level maintained in the asymptomatic patient that may contribute to curtailed disease severity.

We understand that the small cohort size is the limitation of our study. The kinetics of viral persistence, antibody response, and cytokine profile we observed in only 15 COVID-19 patients in our study were parallel to the literature. However, our analysis revealed that even though antibody levels begin to decline two weeks PSO, the neutralization efficacy per unit of antibody remained the same or continued to increase. The process of antibody maturation was delayed in patients with virus persistence. This indicates that the population of antibodies with poor neutralization capacity in the early stages may be deleterious instead of helpful. Timing of plasma procurement can be a critical factor for convalescent plasma therapy. Our results also suggest management of proinflammatory cytokines other than IL-6 may help toward recovery from severe COVID-19, as evidenced by the consolidated benefit of low-dose corticosteroid in treatment [26]. 

## 4. Methods

### 4.1. Patients, Sample Collection and Handling, Biobanking, and Ethics Statement

All COVID-19 patients were RT-PCR-confirmed and placed in negative pressure isolation rooms in our hospital. Nasopharyngeal or oropharyngeal throat swab specimens were collected on stated days post symptom onset for serial RT-PCR tracking. Pharyngeal specimens were also used to isolate and culture SARS-CoV-2 virus strains. One virus strain was used for a neutralization antibody test in our BSL-3 facility. Antibody tests were carried out using serum samples. Peripheral blood was collected for routine medical tests, and serum samples remaining after routine medical tests were used in our research.

Isolated virus strains are deposited in our institutional depository. Sequences of the virus strains are also deposited in the depository of the Taiwan Centers for Disease Control (Taiwan CDC).

This research was performed with informed consent from patients or their families. Specimen sampling and transportation were handled according to the criteria of the Taiwan CDC. This study was approved by the Institutional Review Board of Chang Gung Medical Foundation, Linkou Medical Center, Taoyuan City, Taiwan.

### 4.2. SARS-CoV-2 Nucleic Acid Detection

Nasopharyngeal or oropharyngeal throat swab specimens were collected from patients. Test for SARS-CoV-2 nucleic acid followed standard protocols. RNA was extracted from clinical samples with the LabTurbo system (Taigen, Taiwan). A 25 μL reaction contained 5 μL of RNA, 12.5 μL of 2 × reaction buffer provided with the Superscript III one-step RT-PCR system with Platinum Taq Polymerase (AgPath-ID One-step RT-PCR Kit), 1 μL of reverse transcriptase/Taq mixture from the kit, 0.4 μL of a 50 mM magnesium sulfate solution (Invitrogen), and 1 μg of nonacetylated bovine serum albumin (Roche). All oligonucleotides were synthesized and provided by Tib-Molbiol (Berlin, Germany). Thermal cycling was performed at 48 °C for 30 min for reverse transcription, followed by 95 °C for 10 min and then 45 cycles of 95 °C for 10 s, 65 °C for the 30 s [27].

### 4.3. COVID-19 Serum Antibody Detection

To evaluate the antibody response, the levels of total IgG in patients’ sera were semi-quantified by ELISA (cat No. WS-1096, WANTAI SARS-CoV-2 Ab ELISA, China) through the use of a Triturus ELISA processor, following manufacturer’s instructions. WANTAI SARS-CoV-2 Ab ELISA is a two-step incubation antigen “sandwich” enzyme immunoassay kit, which uses polystyrene microwell strips pre-coated with recombinant SARS-CoV-2 antigen. The results had previously been verified with another ELISA kit (Anti-SARS-CoV-2 ELISA IgG, Euroimmun, Germany), and the ELISA was done in accordance with the manufacturer’s instructions.

### 4.4. Neutralization Antibody Test (NAT)

The neutralizing antibody test of COVID-19 followed the standard protocol of a plaque reduction neutralization test. Vero cells were regularly maintained in minimal essential medium (MEM) supplemented with 10% (*v*/*v*) fetal bovine serum (FBS). COVID-19 virus was propagated in Vero cells in a maintenance medium consisting of MEM supplemented with 0% FBS. Serum samples were inactivated at 56 °C for 30 min before use. Serial two-fold dilutions of sera were mixed with an equal volume of COVID-19 virus suspension containing 100 × the median tissue culture infectious dose (TCID_50_). The mixture was incubated for 2 h at 37 °C, and then an equal volume of suspended VeroE6 cells (approximately 30,000 cells/well) was added to each well. Following incubation for 1 week at 37 °C, cells were fixed with 5% glutaraldehyde and stained with 0.1% crystal violet. Serum neutralization titers were calculated and expressed as the reciprocals of the highest serum dilution that inhibits cytopathic effects.

### 4.5. COVID-19 Serum Cytokine and Chemokine Detection

To evaluate the response of cytokines, chemokines, and other immune molecules, sera were quantified by Bio-Plex Pro Human Cytokine 48-Plex Screening kit (Bio-Rad), as per the manufacturer’s instructions. 

## Figures and Tables

**Figure 1 pathogens-10-00752-f001:**
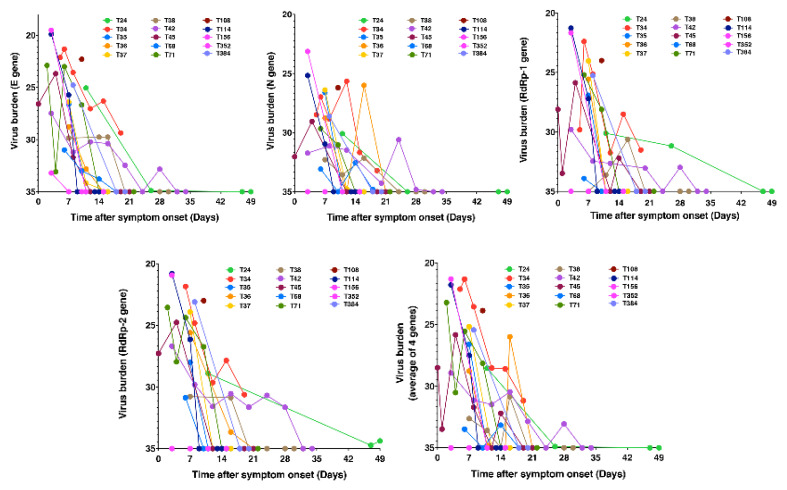
*Pharyngeal virus load and clearance*. Kinetics of virus burden in terms of E, N, RdRp-1, and RdRp-2 gene Ct values at stated time points after symptom onset. Pharyngeal samples were collected and RT-PCR tests were performed as described in the text. (The patients are tagged by our national serial number of COVID-19 cases).

**Figure 2 pathogens-10-00752-f002:**
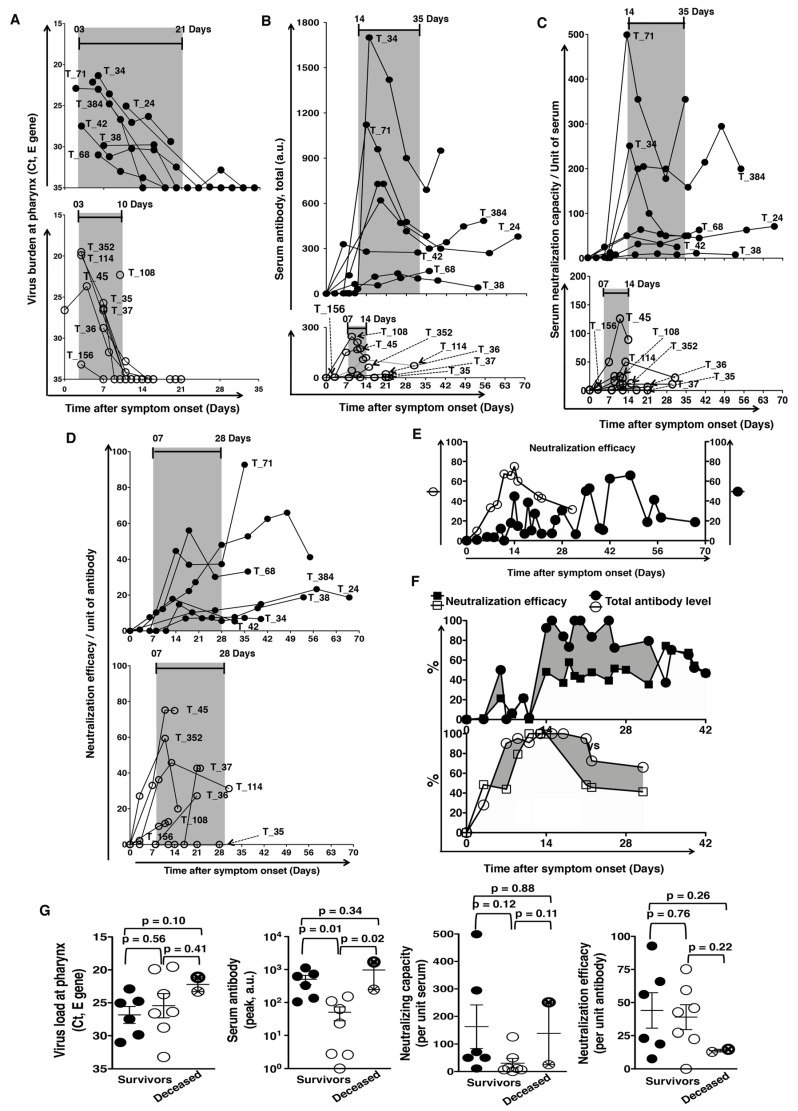
(**A**) *Pharyngeal virus load and clearance.* Of the participants involved, 7 patients revealed delayed clearance of virus (upper panel), and the other 8 patients eradicated the virus quickly (lower panel). (**B**,**C**) *Serum antibody level and neutralization titer.* (**B**) High antibody level per unit of serum and (**C**) proportionally high neutralization titer per unit of serum with persistent presence of the virus (upper panels), compared to those in the patients with rapid virus clearance (lower panels). (**D**,**F**) *Neutralization efficacy per unit of antibody*. (**D**) Comparable neutralization efficacy per unit of antibody between patients with viral persistence (upper panel) and patients with rapid eradication (lower panel). (**E**). Longer average time for peak neutralization efficacy per unit of antibody in patients with viral persistence (closed circle) than those who cleared virus rapidly (open circle). (**F**) There were many more antibodies of compromised neutralization efficacy before the time of peak efficacy (shaded area) in patients with delayed clearance (upper panel), compared to those in patients with rapid clearance (lower panel). (**G**) *Viral load and antibody response in the deceased.* The deceased (the two crossed circles) had higher viral loads on presentation, higher amount of antibody and higher neutralization capacity in unit serum, but they had lower neutralization efficacy per unit of antibody, compared to those who survived (non-crossed circles). (The patients are tagged by our national serial number of COVID-19 cases).

**Figure 3 pathogens-10-00752-f003:**
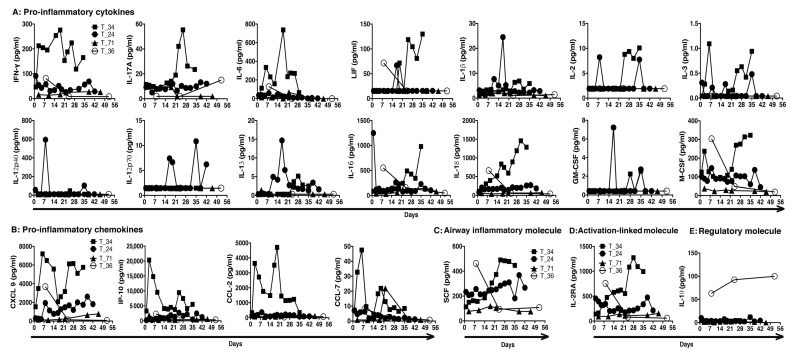
Cytokine profiles of RT-PCR-confirmed COVID-19 patients. Kinetics of (**A**) pro-inflammatory cytokines, (**B**) pro-inflammatory chemokines, (**C**) airway inflammatory molecule SCF, (**D**) immune activation-associated molecule IL-2RA, and (**E**) regulatory molecule IL-10 in the serial sera samples from stated patients during their disease course. Sera were collected from RT-PCR-confirmed COVID-19 patients. (The patients are tagged by our national serial number of COVID-19 cases).

**Table 1 pathogens-10-00752-t001:** Demographic and clinical characteristics of the patients with COVID-19.

Patient Cohort	With Virus Persistence (Mean ± SD)	With Rapid Virus Clearance (Mean ± SD)
Number	7 (2 Male/5 Female)	8 (3 Male/5 Female)
Age (Years)	60.14 ± 3.58	38.25 ± 5.28
ICU assistance	3/7	1/8
ECMO support	2/7	1/8
Mortality	1/7 (14.3%)	1/8 (12.5%)
Virus burden at presentation:		
E gene, Ct value	26.02 ± 3.59	25.75 ± 2.67
RdRp1 gene, Ct value	27.87 ± 5.68	25.88 ± 4.53
RdRp2 gene, Ct value	26.35 ± 5.27	25.61 ± 4.73
N gene, Ct value	28.52 ± 5.34	27.99 ± 4.04
Virus clearance		
Days, PSO	24.14 ± 4.33	10.25 ± 0.56
Anti-spike IgG antibody Response in sera:		
Peak levels	677.2 ± 217.8	76.70 ± 32.11
Time for peak (Days, PSO)	17.43 ± 2.61	11.13 ± 2.48
Virus-neutralizing antibodies		
Capacity/unit sera (Peak levels)	168.00 ± 63.42	29.68 ± 14.82
Time for peak (Days, PSO)	30.43 ± 7.80	12.20 ± 5.17
Efficacy/unit antibody (Peak levels)	41.37 ± 11.49	35.70 ± 8.78
Time for peak (Days, PSO)	34.14 ± 7.15	12.50 ± 2.35

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
