# Peer review of "Relative COVID-19 Viral Persistence and Antibody Kinetics"

_pathogens, 2021, doi:10.3390/pathogens10060752_

Round 1

Reviewer 1 Report

The authors have added Bio-Rad 48-plex cytokine screening kit results to the previous dataset. Additionally, Table 1 was added to clarify the classification into two groups (table title indicates 47 patients, when there in fact were 15) in which some severity-indicating data (ECMO support, ICU assistance) were also included. More references have been added, and some sections have edited according to my suggestions.

Overall, the manuscript still describes with a very small number of COVID-19 patients (n=15) viral RNA, antibody and cytokine kinetics within 2-3 first months. As previously stated these kinetics, as well as cytokine profiles and neutralization efficacy, have been previously far more extensively and with better quality of presentation reported by numerous publications with larger sample sets, longer follow-up and broader testing. With its low number of patients and rather unclearly presented short, narrow follow-up data, I do not see this manuscript adding any useful information to the existing body of literature. As it appears the authors cannot overcome this, I also refrain from commenting the other major flaws in the manuscript (conclusions, clarity of presentation, etc.). I recommend against publication of this manuscript in Pathogens. 

Author Response

Reviewer #1:

The authors have added Bio-Rad 48-plex cytokine screening kit results to the previous dataset. Additionally, Table 1 was added to clarify the classification into two groups (table title indicates 47 patients, when there in fact were 15) in which some severity-indicating data (ECMO support, ICU assistance) were also included. More references have been added, and some sections have edited according to my suggestions.

Answer:We appreciate your comments and concerns.

Overall, the manuscript still describes with a very small number of COVID-19 patients (n=15) viral RNA, antibody and cytokine kinetics within 2-3 first months. As previously stated these kinetics, as well as cytokine profiles and neutralization efficacy, have been previously far more extensively and with better quality of presentation reported by numerous publications with larger sample sets, longer follow-up and broader testing. With its low number of patients and rather unclearly presented short, narrow follow-up data, I do not see this manuscript adding any useful information to the existing body of literature. As it appears the authors cannot overcome this, I also refrain from commenting the other major flaws in the manuscript (conclusions, clarity of presentation, etc.). I recommend against publication of this manuscript in Pathogens.

Answer:We appreciate your time and effort for reviewing our work.

Reviewer 2 Report

This report includes data from 15 patients with SARS-CoV-2 infection diagnosed by E-gene RT-PCR. The 15 cases are divided into two groups: short persistence of virus (n=7) or over 2 weeks persistence (n=8). Patients with more than 2 weeks clearance are much older than the group with short persistence.  The older patients also show later serum anti-spike IgG ELISA and NT titer peaks. 

It is not stated how or if the patients consented to participate. This information should be included.

In table 1 it is stated that there are 47 patients, but data are only reported from 15 participants. Where are the data from the remaining 32 patients? 

This study would be more worthy for publishing if the authors include data from the rest of the 47 included patients.

In the method section, a description of the time points for sampling should be described. The types of and number of different specimens should also be mentioned. Biobanking and handling of biological material should be described. 

With regard to both title and text in 4.1, the nucleic acid detection description should be changed to "SARS-CoV-2 nucleic acid detection",  as it is not a test for detection of the illness COVID-19.

In the result section, the virological data from the different time point should be showed in a graph or table with ct-values from the PCR tests over time.

In the discussion, the authors should include a brief explanation of the main results in the first paragraph, before discussing their findings. The discussion should be more focused towards the study findings, it is now too general. More detailed discussion on the antibody differences between the two groups must be included as well as addressing how these differences could be explained by the large age discrepancy between the groups.  The authors should also state that the study´s major limitation is the limited study population with very few patients included.

The abstract needs major revision. The results should be reported with number of participants and main findings. It is also lacking conclusions. In the abstract there is a sentence about cytokines in the deceased patient. In the results/table 1, it seems that there were two deceased patients, one in each group. This should be corrected in the abstract.

Author Response

Reviewer #2:

This report includes data from 15 patients with SARS-CoV-2 infection diagnosed by E-gene RT-PCR. The 15 cases are divided into two groups: short persistence of virus (n=7) or over 2 weeks persistence (n=8). Patients with more than 2 weeks clearance are much older than the group with short persistence. The older patients also show later serum anti-spike IgG ELISA and NT titer peaks. It is not stated how or if the patients consented to participate. This information should be included.

Answer: We took consent from patients or their families. It is mentioned in the Introduction part of revised manuscript as “With informed consent from patients or their families, our research was conducted using serum samples remaining after routine medical tests” in the introduction part. We also added a section on patients and their consents, sample collection and handling, and ethics statement in the methods section of the revised manuscript, as suggested.

In table 1 it is stated that there are 47 patients, but data are only reported from 15 participants. Where are the data from the remaining 32 patients? This study would be more worthy for publishing if the authors include data from the rest of the 47 included patients.

Answer: We apologize for the typo “47” in Table 1 title. Our study involves 15 participants, and we have logistic problems in getting in more patients and doing longer followup due to a variety of reasons including the local epidemiology of COVID-19 infection.

In the method section, a description of the time points for sampling should be described. The types of and number of different specimens should also be mentioned. Biobanking and handling of biological material should be described.

Answer: All the time points for sampling are the time points after symptoms onset. We made it clear in the revised manuscript. We also added a section on patients and their consents, sample collection and handling, and biobanking in the methods section of the revised manuscript, as suggested.

With regard to both title and text in 4.1, the nucleic acid detection description should be changed to "SARS-CoV-2 nucleic acid detection", as it is not a test for detection of the illness COVID-19.

Answer: We amended the text, as suggested.

In the result section, the virological data from the different time point should be showed in a graph or table with ctvalues from the PCR tests over time.

Answer: We provided the virological data of each patient as a new Figure (Figure 1) in the revised manuscript, as suggested.

In the discussion, the authors should include a brief explanation of the main results in the first paragraph, before discussing their findings. The discussion should be more focused towards the study findings, it is now too general. More detailed discussion on the antibody differences between the two groups must be included as well as addressing how these differences could be

explained by the large age discrepancy between the groups. The authors should also state that the study´s major limitation is the limited study population with very few patients included.

Answer: We amended the discussion in the revised manuscript, as suggested.

The abstract needs major revision. The results should be reported with number of participants and main findings. It is also lacking conclusions. In the abstract there is a sentence about cytokines in the deceased patient. In the results/table 1, it seems that there were two deceased

patients, one in each group. This should be corrected in the abstract.

Answer: We amended the abstract in the revised manuscript, as suggested.

Round 2

Reviewer 1 Report

No new changes have been indicated. I wish the authors best of luck in their future efforts.

Author Response

No new changes have been indicated. I wish the authors best of luck in their future efforts

Answer: We appreciate your time and effort in reviewing our work.

Reviewer 2 Report

Recommend a check with English language expert before publishing.

Also, when reading the revised manuscript I found a wording that needs change: "harmful antibodies". It is not the antibodies as such that is the problem with SARS, so I suggest to omit "harmful".

Author Response

English language and style are fine/minor spell check required.

Answer:We checked spellings and amended the revised manuscript, as suggested.

Recommend a check with English language expert before publishing. Also, when reading the revised manuscript I found a wording that needs change: "harmful antibodies". It is not the antibodies as such that is the problem with SARS, so I suggest to omit "harmful".

Answer: We took further help for the English language and omitted "harmful" from the "harmful antibodies", as suggested.

This manuscript is a resubmission of an earlier submission. The following is a list of the peer review reports and author responses from that submission.

Round 1

Reviewer 1 Report

The authors present data from 15 patients with SARS-CoV-2 infection diagnosed by RT-PCR. The patients are divided into two groups by whether viral RNA in pharyngeal swabs persists less (n=7) or more (n=8) than two weeks. Patients with slower clearance are older, and have higher and later serum anti-spike IgG ELISA and PRNT titer peaks. Such a division is not established in the literature and the authors provide little justification for it. Overall, the fast and slow clearance group characteristics resemble those of mild and severe disease. Regrettably, no data are presented on disease severity for any of the individuals, with the exception of two deceased individuals with high initial viral loads and antibody titers, for whom it remains unclear whether they belong to the fast or slow clearance group. The methods are appropriate.

Overall, the manuscript describes with a very small number of individuals (n=15) viral RNA and antibody kinetics of SARS-CoV-2 in first three months. These kinetics have already previously been extensively documented by numerous publications (see below) with larger sample sets with longer follow-up as well as broader testing, e.g. different sampling sites, antibodies against different antigens.

Thus, I regrettably do not consider the manuscript in its current form to 1) add significant information to the existing literature or 2) be of interest to the readers of Pathogens. To improve the manuscript to fulfill these criteria, e.g. more individuals could be included, ideally with a longer follow-up and clinical data (at least on disease severity) included.

Due to these major issues I for now refrain from extensive commenting of the details. However, some minor comments: Discussion on lines 81-117 is too long, with obvious/irrelevant description of e.g. affinity maturation. Lines 118-119 refer to HCWs in the sample set, however, there is no mention of HCWs elsewhere, so it remains unknown to what data these lines refer to. Sandwich ELISA is a standard method, and it is unnecessary to describe its principle in the methods (lines 141-156), on the other hand e.g. the machine that was used to measure the absorbances should be mentioned.

References

Sun J, Xiao J, Sun R, et al. Prolonged Persistence of SARS-CoV-2 RNA in Body Fluids. Emerg Infect Dis. 2020;26(8):1834-1838. https://doi.org/10.3201/eid2608.201097

Carmo, A, Pereira‐Vaz, J, Mota, V, et al. Clearance and persistence of SARS‐CoV‐2 RNA in patients with COVID‐19. J Med Virol. 2020; 92: 2227– 2231. https://doi.org/10.1002/jmv.26103

Lau EHY, Tsang OTY, Hui DSC, et al. Neutralizing antibody titres in SARS-CoV-2 infections. Nat Commun. 2021;12(1):63. Published 2021 Jan 4. https://doi.org/10.1038/s41467-020-20247-4

Marklund E, Leach S, Axelsson H, et al. Serum-IgG responses to SARS-CoV-2 after mild and severe COVID-19 infection and analysis of IgG non-responders. PLoS One. 2020;15(10):e0241104. Published 2020 Oct 21. https://doi.org/10.1371/journal.pone.0241104

Seow J, Graham C, Merrick B, et al. Longitudinal observation and decline of neutralizing antibody responses in the three months following SARS-CoV-2 infection in humans. Nat Microbiol. 2020;5(12):1598-1607. https://doi.org/10.1038/s41564-020-00813-8

Zhang X, Lu S, Li H, et al. Viral and Antibody Kinetics of COVID-19 Patients with Different Disease Severities in Acute and Convalescent Phases: A 6-Month Follow-Up Study. Virol Sin. 2020;35(6):820-829. https://doi.org/10.1007/s12250-020-00329-9

Sun J, Tang X, Bai R, et al. The kinetics of viral load and antibodies to SARS-CoV-2. Clin Microbiol Infect. 2020;26(12):1690.e1-1690.e4. https://doi.org/10.1016/j.cmi.2020.08.043

Wang Y, Zhang L, Sang L, et al. Kinetics of viral load and antibody response in relation to COVID-19 severity. J Clin Invest. 2020;130(10):5235-5244. https://doi.org/10.1172/JCI138759

Reviewer 2 Report

Small cohort study including only 15 patients, although the virological characterization is thorough. Virus culture would add value in order to address the question of contagiousness of patients with long lasting positive PCR tests. Suggest to expand the cohort with more patients to improve the quality of the study. Many similar studies have now been published with more than ten times the number of patients, thereby generating more reliable results. It is not stated if the patients consented to participate.